# A Nanoscale Sensor Based on a Toroidal Cavity with a Built-In Elliptical Ring Structure for Temperature Sensing Application

**DOI:** 10.3390/nano12193396

**Published:** 2022-09-28

**Authors:** Feng Liu, Shubin Yan, Lifang Shen, Pengwei Liu, Lili Chen, Xiaoyu Zhang, Guang Liu, Jilai Liu, Tingsong Li, Yifeng Ren

**Affiliations:** 1School of Electrical and Control Engineering, North University of China, Taiyuan 030051, China; 2School of Electrical Engineering, Zhejiang University of Water Resources and Electric Power, Hangzhou 310018, China; 3Joint Laboratory of Intelligent Equipment and System for Water Conservancy and Hydropower Safety Monitoring of Zhejiang Province and Belarus, Hangzhou 310018, China

**Keywords:** refractive index sensor, Fano resonance, MIM, temperature sensing

## Abstract

In this article, a refractive index sensor based on Fano resonance, which is generated by the coupling of a metal–insulator–metal (MIM) waveguide structure and a toroidal cavity with a built-in elliptical ring (TCER) structure, is presented. The finite element method (FEM) was employed to analyze the propagation characteristics of the integral structure. The effects of refractive index and different geometric parameters of the structure on the sensing characteristics were evaluated. The maximum sensitivity was 2220 nm/RIU with a figure of merit (FOM) of 58.7, which is the best performance level that the designed structure could achieve. Moreover, due to its high sensitivity and simple structure, the refractive index sensor can be applied in the field of temperature detection, and its sensitivity is calculated to be 1.187 nm/℃.

## 1. Introduction

Surface plasmon polaritons (SPPs) are electromagnetic waves consisting of the coupling of free electron and incident photon at the interface between metal and medium [1]. The electromagnetic wave can be transmitted on the interface and confine most of the field energy to the interface, while its intensity decreases rapidly in the vertical direction [2,3]. Because of the characteristics of SPPs, such as breaking the diffraction limit and strong local characteristics, this technology can be used to manufacture many instruments, such as Wiener sensors, filters, all-optical switches, and optical splitters [4,5]. This is of great significance to the development of light sensing, super-resolution imaging, and information processing and transmission [6]. SPPs can appear at the metal–medium interface; therefore, many metal and medium waveguides can excite SPPs, such as metal–dielectric–metal waveguides [7], medium–metal–dielectric waveguides [8], and medium–dielectric–metal [9]. Among them, MIM waveguides can better limit SPPs, have smaller ohmic losses, and achieve a longer SPP transmission distance with a very small size [10]. Therefore, many coupled cavities based on MIM waveguide structures have been discovered, such as hexagonal, toroidal, rectangular, u-shaped, and semicircular cavities [11,12,13,14].

Many optical phenomena such as Fano resonance have been observed in plasma-waveguide coupling systems [15]. In an atomic system, when the continuous energy band and the discrete energy level interact, the interference effect will occur [16], and the zero-absorption phenomenon will occur at a specific optical frequency, which is called the Fano resonance effect [17,18,19]. In the SPP system, the Fano resonance is generated by the mechanism: when light is incident into the MIM straight waveguide a direct-coupled differential bright-state superradiant mode occurs, with a relatively flat line pattern. However, a part of the metal nanoparticles cannot be directly excited by light and need the coupling of bright mode to be excited, which is called a dark-state sub-radiation mode with a narrower and sharper line pattern. The Fano resonance phenomenon arises when these two radiative states overlap and interfere with each other, which is a sharp and asymmetric curve [20]. Fano resonance is the occurrence of zero absorption at a specific optical frequency, resulting in spectral line asymmetry. Compared with traditional isoplasmon resonance, the transmission spectrum of Fano resonance has a narrower linewidth, is more sensitive to changes in its own structure and surrounding environment, and has stronger local field strength and better sensitivity [21,22,23].

In this study, a nanoscale refractive index sensor structure is raised and investigated, which comprises an MIM waveguide and a toroidal cavity with a built-in elliptical ring. The propagation properties of the designed structure are analyzed, adopting finite element method (FEM) [24] and coupling mode theory (CMT) [25]. The simulation results indicate that the asymmetric Fano resonance waveform is generated by the interference between the continuous wideband mode effected by the downward rectangular baffle on the waveguide and the discrete narrowband mode caused by the TCER structure. Since the spectrum of Fano resonance is easily influenced by the geometric parameters of the designed structure, the effect of geometric parameters on the propagation properties is investigated. These geometric parameters include the minor semi-axis of the ellipse, the outer radius of the TCER, the height of the rectangular baffle, and the coupling gap between the MIM waveguide and the TCER structure.

## 2. Materials and Methods

The elementary diagram of the designed structure is illustrated in Figure 1. The integral structure consists of an MIM waveguide with a rectangular baffle and a toroidal cavity with a built-in elliptical ring. Since the simulation of 3D structures has high requirements for hardware configuration and mesh division, while the magnetic field properties of 3D structures do not seriously diverge from those of 2D structures, the two-dimensional model can be used to approximate the three-dimensional model for calculation. R and r represent the radii of the outer and inner circles of the TCER structure, respectively, and the semi-major axis of the inner and outer ellipse is set to be the same as the radius of the inner and outer circle. Furthermore, d expresses the length of the semi-minor axis of the outer ellipse. The height of the single rectangle baffle is h, and g denotes the coupling distance between the waveguide and the TCER, while w denotes the width of the MIM waveguide and the two annular cavities and rectangular baffle. When the direct coupling distance between the upper and lower intersection is relatively small, the SPP waves transmitted in the upper and lower metal–dielectric interfaces will couple with each other to form two dispersion models, namely the even-symmetric mode and the odd-symmetric mode. In the odd-symmetric mode, the SPPs have a shorter propagation distance and higher energy loss in transmission. In the even-symmetric mode, the transmission distance is long, and the energy loss is light. In the MIM waveguide structure, in order to ensure the existence of only one transmission mode of even symmetry, the width of the dielectric layer is generally set to 50 nm.

The white portion and the yellow portion in Figure 1 represent air and silver, respectively. The relative permittivity $\varepsilon_{d}$ of air is 1, and the relative permittivity of Ag is defined as follows [26]:(1)$$\;\;\;\;\;\varepsilon \left(\omega \right)=\varepsilon_{\infty }+\frac{\varepsilon_{s}-\varepsilon_{\infty }}{1+i\tau \omega }+\frac{\sigma }{i\omega \varepsilon_{0}}$$

The transverse magnetic mode formula of the MIM waveguide is given below [27]:(2)$$\tanh\left(k\omega \right)=-\frac{2k\alpha_{c}}{k^{2}+p^{2}\alpha_{c}}$$
where $p=\varepsilon_{in}/\varepsilon_{m}$, $\alpha_{c}={\left[k_{0}{}^{2}*\left(\varepsilon_{in}-\varepsilon_{m}\right)+k\right]}^{\frac{1}{2}}$, $\varepsilon_{in}$ and $\varepsilon_{m}$ denote the permittivity of dielectric and metal, respectively; *k* represents the wave vector in the waveguide, and *k* can be described as $k_{0}=2\pi /\lambda_{0}$, in free space.

In this paper, we propose three parameters to evaluate the characteristics of the sensor, they are FWHM, sensitivity (S), and figure of merit (FOM). FWHM is applied to define the sharpness of the spectral line, which refers to the width of the spectral line at the half height of the formant. FOM can comprehensively evaluate the two parameters of sensitivity and FWHM. S and FOM are, respectively, expressed by the following formulas [28,29]:(3)S=∆λ/∆n
(4)FOM=S/FWHM
where ∆λ and ∆n represent the variation of refractive index and wavelength, respectively.

COMSOL Multiphysics 5.4a was used to establish the geometric model of the designed sensor system. A perfect match layer was established to absorb light reflected to the outside. Hyper-detailed triangles were selected for mesh dissection in the waveguide and TCER structure regions, and regular triangles were selected for mesh dissection in other regions, which can reduce unnecessary computation while ensuring computational accuracy. Then, simulations were performed for the wavelength range 1400–2200 in steps of 1 nm.

## 3. Simulation Results and Analysis

The sensing performances of a single annular cavity and the TCER structure were investigated separately. After comparison, it was found that the sensitivity of the TCER structure was slightly higher than that of the single-ring structure within the set refractive index variation range, while the TCER structure had a higher FOM value, so TCER was chosen for further study.

The initial structural parameters of the sensor were defined and configured as follows: R = 240 nm, d = 150 nm, h = 80 nm, g = 10 nm. In order to better understand the Fano resonance generation process and the propagation properties of the integral structure, the single TCER structure and the single rectangle baffle structure were simulated, respectively, as illustrated in Figure 2. The black, red, and blue lines represent the transmission spectra of the rectangle baffle structure, the TCER structure, and the entire system, respectively. In the figure, it can be noticed that the transmission spectrum of the entire system has an asymmetric sharp shape, i.e., Fano resonance, which is generated by the interference between the continuous broadband mode and the discrete narrowband mode. The transmission spectrum of a single rectangular baffle structure is a slightly upward sloping straight line, and the points on the line have high transmittance, which is regarded as a continuous broadband mode. The transmission spectrum of the straight waveguide coupled TCER structure has a symmetric trough, which is a typical Lorentz resonance and can be regarded as the discrete narrowband mode that generates the Fano resonance.

In order to better explain the generation process of Fano resonance, the magnetic field distributions of the TCER structure and the entire system at the resonance dip (λ = 1891 nm) were investigated, which is showed in Figure 3a,b, respectively. It can be found that SPPs in both structures can pass through the waveguide and be coupled to the TCER. According to Figure 3b, the normalized magnetic field is mainly distributed in the single TCER structure, and a small part is in the bus waveguide, indicating that obvious resonance occurs. In addition, the upper and lower parts of the TCER are antiphase. The single TCER structure has a similar normalized magnetic field distribution to the whole structure, but it has more magnetic field distribution at the waveguide. Therefore, the addition of a rectangular baffle in the waveguide can reduce the propagation of SPPs in the waveguide to generate stronger resonance in the TCER and promote the generation of Fano resonance.

The effect of refractive index variation on the sensing performance of the designed structure was investigated. On the basis of constant structure parameters, R = 240 nm, d = 150 nm, h = 80 nm, w = 50 nm, g = 10 nm, the refractive indexes n = 1.00, n = 1.01, n = 1.02, n = 1.03, n = 1.04 and n = 1.05 RIU were respectively simulated, and the results are illustrated in the Figure 4a,b. Based on Figure 4a, as the refractive index rises, the transmission spectrum shows an almost isometric redshift. Refractive index variation will not affect the waveform of transmission spectrum but only alter the position of transmission spectrum. As shown in Figure 4b, the wavelength shift of the dip will change linearly with the variation of refractive indices. Based on this feature, the proposed structure can be utilized as a refractive index sensor. The sensitivity of the sensor was 2220 nm/RIU by calculating the slope of the fitted line, and the optimal value was calculated as 58.4, which is the best parameter of the structure.

Since the Fano resonance is greatly affected by the structural geometric parameters, we investigated in detail the effect of different parameters on the transmission spectrum of the Fano resonance. To start with, the influence of the outer circle radius of the TCER structure on Fano resonance was studied. The outer radius R was adapted from 220 to 240 nm at 5 nm separation, with other parameters kept invariant. Figure 5a shows the transmission spectra generated at various external radii; the transmittance at the dip increased slightly as R increased. Moreover, with the increase of radius R, the curve had an obvious red shift, which indicates that radius R is a key parameter, determining the trough wavelength of Fano resonance. The sensitivity of different radii was obtained by linear fitting, which is illustrated in Figure 5b. It can be perceived from the figure that the refractive index increases from 1720 to 2220 nm/RIU as the TCER structure outer radius increases. Therefore, in practical application, the appropriate radius can be selected according to the sensitivity requirements.

Then, the effect of the external ellipse minor semi-axis, another structural parameter of the TCER, on the propagation performance of the sensor was investigated. The minor semi-axis d was adapted from 110 to 150 nm at 10 nm intervals, with other parameters unchanged. The simulation results and the sensitivity fitting line are plotted in Figure 6a,b, separately. A slight redshift of the transmission spectrum can be noticed with the growth of d. However, the shape of the transmission spectrum and the FWHM were basically unchanged. As can be seen from Figure 6b, the sensitivity of the whole system can be slightly improved when the minor semi-axis becomes larger. Therefore, the length of the minor axis can be changed appropriately according to the structural requirements without affecting the sensitivity greatly.

Subsequently, the influence of the structural parameters of the MIM waveguide on the transmission characteristics of the sensor was evaluated. The height of the rectangular baffle was set as 40, 50, 60, 70, and 80 nm, while other parameters were kept invariant. As illustrated in the Figure 7, the dip position of the Fano resonance remained invariant no matter how h varied, while the asymmetry degree of the Fano line shape increased gradually with the increase of h. In the process of Fano resonance generation, the MIM waveguide generates a continuous broadband. From the above analysis, it can be concluded that the continuous broadband state has an influence on the line profile of Fano resonance but not on the wavelength of the dip.

Finally, the influence of the coupling gap between the whole system and the TCER structure on transmission characteristics was investigated. The g was adjusted from 5 to 25 nm, with other parameters kept constant. As represented in Figure 8a, the curves had an obvious blue shift, and the transmittance gradually increased with the increase of the coupling gap. Moreover, it can be found that the FWHM becomes significantly narrower as g increases from 5 to 25 nm in Figure 8b. It can be concluded that the coupling degree of the SPPs and the TCER structure becomes weaker with the increase of g. The fitting lines of sensitivity under various coupling gaps are shown in Figure 8c. The sensitivity decreased with the increase of the coupling gap and reached the maximum of 2500 nm/RIU when g = 5 nm. However, the value of FWHM at g = 5 nm was much larger than that at g = 10 nm, resulting in a relatively small value of figure of merit (FOM); therefore, the sensing performance of the system reached the optimal when g = 10 nm. At this time, the sensitivity was 2220 nm/RIU and FOM was 58.7, which is better than many existing parameters listed in Table 1.

## 4. Application in Temperature Sensing

The above structure has a relatively high sensitivity and is easy to integrate, so it can be applied in the field of temperature sensing. The temperature sensing medium ethanol was chosen as a filler material for the TCER structures, waveguides, and rectangular baffles due to its superior refractive index temperature parameters of $3.94\times 10^{-4}\;\left({℃}^{-1}\right)$. The refractive index temperature parameters of silver and quartz are $9.30\times 10^{-6}\;\left({℃}^{-1}\right)$ and $8.60\times 10^{-6}\;\left({℃}^{-1}\right)$, respectively, which are much smaller than those of ethanol, so the effect of temperature on the refractive index of silver and quartz can be neglected. The refractive index of ethanol is linearly dependent on temperature within the range of melting point of −144 ℃ and boiling point of 78 ℃, which can be expressed as follows [33]:(5)$$n=1.36048-3.94\times 10^{-4}\left(T-T_{0}\right)$$
where *T* denotes the ambient temperature, and $T_{0}$ indicates that the room temperature is set to 20 °C.

In consideration of the melting and boiling points of ethanol, the working temperature range of the temperature sensor was set from −80 °C to 70 °C in order to obtain better operational performance. Ethanol has a melting point of −144 °C, and it is expected that the lowest temperature that the sensor can actually measure is −130 °C. However, if the temperature is below −80 °C or above 70 °C, it will reduce the accuracy of the sensor measurement and increase the measurement error. The structure parameters were set as follows: R = 240 nm, d = 150 nm, h = 80 nm, w = 50 nm, and g = 10 nm. The refractive index of the temperature sensing medium is sensitive to the temperature variations. Therefore, the working principle of the temperature sensor is to calculate the refractive index change of ethanol through the transmission spectrum displacement so as to achieve the purpose of temperature detection. The sensitivity $S_{T}$ of the temperature sensor can be calculated using the following formula:(6)$$S_{T}=∆\lambda /∆T$$
where ∆*λ* represents the translation of transmission spectrum, and ∆*T* represents the temperature change, which is set to 150 °C according to the test temperature range.

When the temperature rises from −80 °C to 70 °C in 30 °C increments, the test results are shown in the Figure 9 above. It can be seen from Figure 9a that the transmission spectrum shows a significant blue shift with increasing temperature. The accuracy of modern optical instruments can reach the nanometer level and accurately detect the wavelength. A FWHM = 202 nm can be obtained, and the trough position is shifted from 2774 nm to 2952 nm; thus, we can derive ∆*λ* = 178 nm. Figure 9b shows the sensitivity fitting line of the sensor, by linear fitting, the accuracy of the measurement can be guaranteed. The sensitivity of the temperature sensor has been calculated to be 1.187 nm/°C and the FOM = 0.00587.

## 5. Conclusions

In this study, a nanoscale refractive index sensor based on Fano resonance, which is generated by the coupling of an MIM waveguide structure and a TCER structure, was designed, and the propagation properties of the integral structure were analyzed by adopting FEM. Then, we analyzed the effects of refractive index and geometric parameters of TCER and MIM waveguides on the sensing characteristics of the integral structure. The wavelength of Fano resonance dip had a redshift with the increase of the refractive index n, the radius of the outer circle R, and the semi-minor axis d of the outer ellipse. However, with the increment of the coupling gap, the wavelength of Fano resonance dip displayed a significant blue shift. The height variation of the rectangular baffle had an influence on the shape of the Fano resonance curve and had no effect on the wavelength of the trough. When the structure parameters were R = 240 nm, d = 150 nm, h = 80 nm, w = 50 nm, and g = 10 nm, the sensing performance of the structure reached its best. The sensor has a maximum sensitivity of 2220 nm/RIU and an FOM of 58.7. Finally, the application of this structure in the field of temperature sensing was studied, and the sensitivity of temperature sensors based on it can reach 1.187 nm/°C. In addition, this design structure also has good prospective applications in other optical fields.

## Figures and Tables

**Figure 1 nanomaterials-12-03396-f001:**
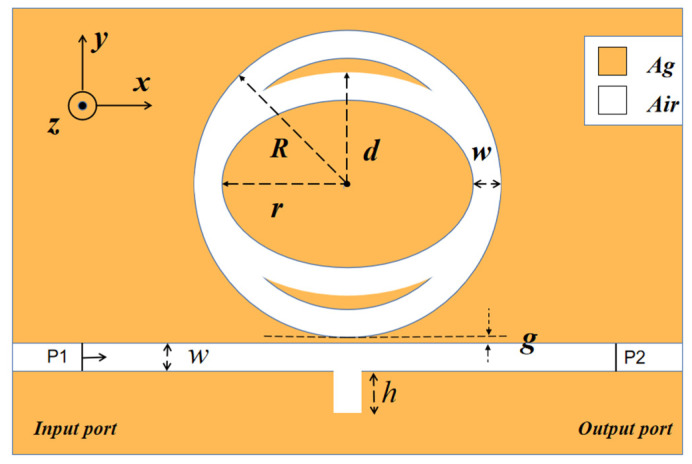
2D schematic of an MIM waveguide and a toroidal cavity with a built-in elliptical ring (TCER).

**Figure 2 nanomaterials-12-03396-f002:**
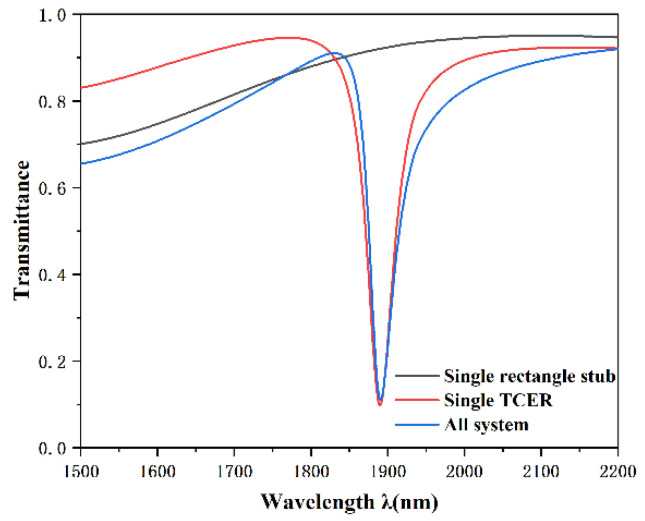
Transmission spectra of the rectangle stub (black line), the TCER structure (red line), and integral structure (blue line).

**Figure 3 nanomaterials-12-03396-f003:**
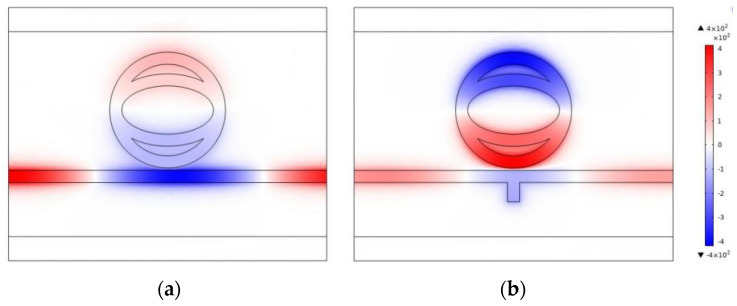
Normalized magnetic field distributions at λ = 1891 nm of (**a**) the TCER structure; (**b**) the entire system.

**Figure 4 nanomaterials-12-03396-f004:**
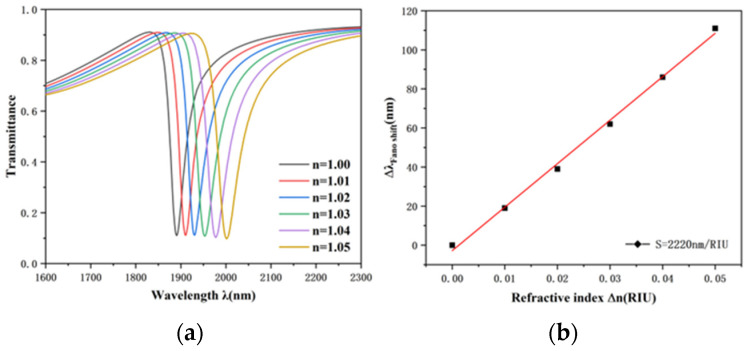
(**a**) Transmission spectra for diverse refractive indices. (**b**) Fitting line of dip wavelength change with refractive index variation.

**Figure 5 nanomaterials-12-03396-f005:**
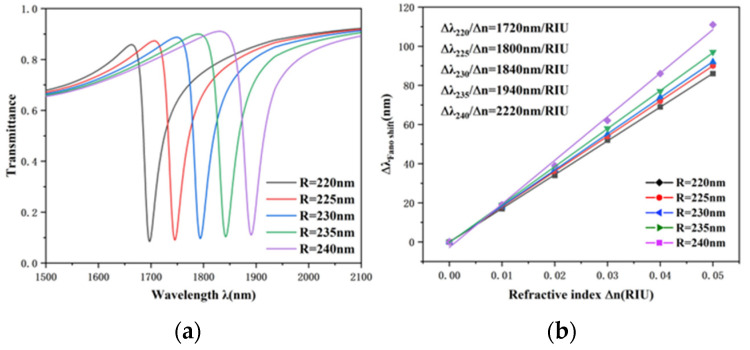
(**a**) Transmission spectra of the entire structure at various radii R. (**b**) Fitting line of sensitivity under diverse radii.

**Figure 6 nanomaterials-12-03396-f006:**
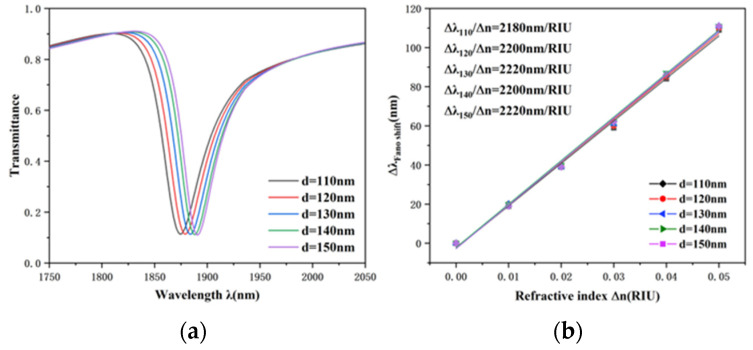
(**a**) Transmission spectra of TCER structure under disparate minor semi-axes of outer ellipses. (**b**) Fitting line of sensitivity at diverse minor semi-axes.

**Figure 7 nanomaterials-12-03396-f007:**
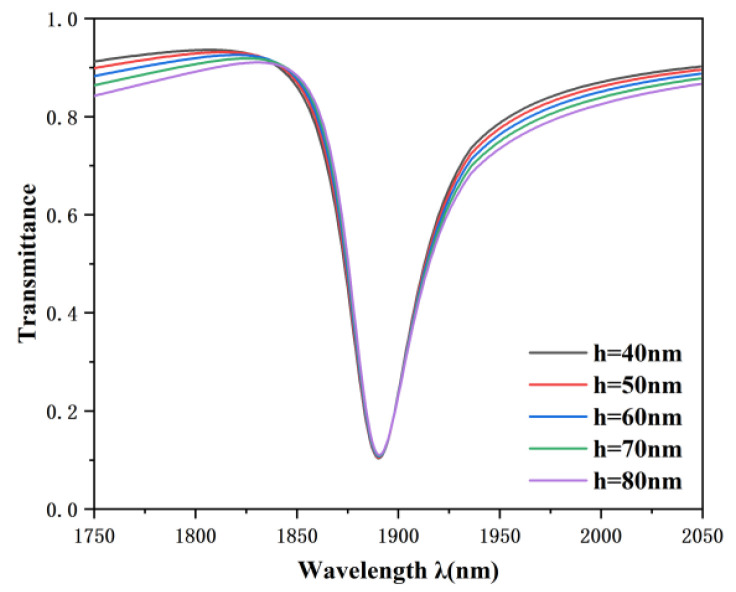
Transmission spectra at various rectangular baffle heights.

**Figure 8 nanomaterials-12-03396-f008:**
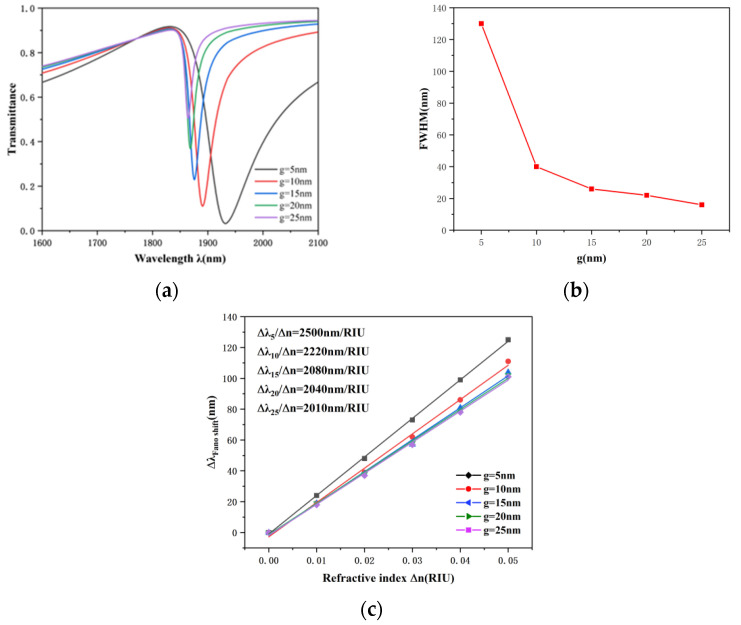
(**a**) Transmission spectra under different coupling gaps. (**b**) Changes of FWHM values at different coupling distances. (**c**) Fitting line of sensitivity at diverse coupling gaps.

**Figure 9 nanomaterials-12-03396-f009:**
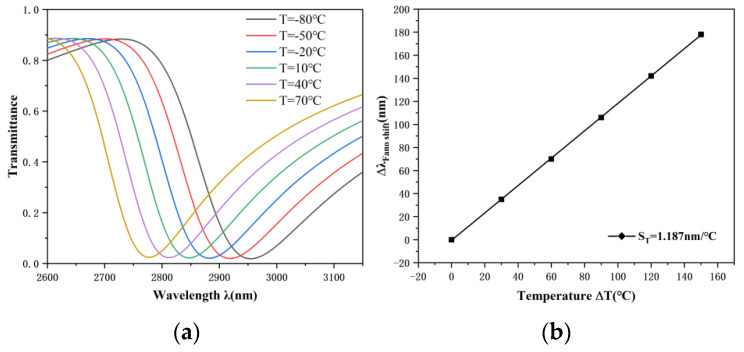
(**a**) Transmission spectra at gradually increasing temperature. (**b**) Sensitivity fitting line for temperature sensors.

**Table 1 nanomaterials-12-03396-t001:** Comparison with data from other literature.

References	Structure Type	Sensitivity (nm/RIU)	Figure of Merit
[30]	Ring splitting cavity	1200	122
[31]	Hexagonal cavity	1562.5	38.6
[32]	Split ring resonator with rectangular short column	1200	40
This work	TCER structure	2220	58.7

## Data Availability

The data presented in this study are available on request from the corresponding author.

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
