# Peer review of "A Nanoscale Sensor Based on a Toroidal Cavity with a Built-In Elliptical Ring Structure for Temperature Sensing Application"

_nanomaterials, 2022, doi:10.3390/nano12193396_

Round 1
Reviewer 1 Report
In the paper 'A nanoscale sensor based on a toroidal cavity with built-in elliptical ring structure for temperature sensing application' the authors present their recent results related to the design of a temperature sensors. In general the paper is well written and the results are clearly presented with the necessary details for the reader. However, I have few concerns which I list in the following:
1. I feel that in this paper (and in particular in the introduction) is completely missing the concept of Fano resonances in dielectric resonators and all the aspect related to the excitations of the anapole mode in such kind of structures. Few important references about this issue are, for instance: Tuning the second-harmonic generation in AlGaAs nanodimers via non-radiative state optimization, Metal–dielectric hybrid nanoantennas for efficient frequency conversion at the anapole mode, Doubly mirror-induced electric and magnetic anapole modes in metal-dielectric-metal nanoresonators, Seeing the unseen: experimental observation of magnetic anapole state inside a high index dielectric particle, Nonradiating anapole modes in dielectric nanoparticles, Generalized hybrid anapole modes in all-dielectric ellipsoid particles
2. I suggest to increase the size letter in figure 1. Also, black letter and blue background is not readable solution.
3. Line 213 remove '..'. Line 22 remove '_0'. Line 231 remove '_T'. Line 246 remove '..'.
4. The authors calculate a sensitivity of 1.187 nm/°C. I think the authors should better explain if it is easy to experimentally detect such small variations? Also, which is the minimum temperature that the authors expect to be realistically measuread by the proposed structure?
Reviewer 2 Report
This paper contains multiple imperfections and must undergo the significant improvements. Some of them are listed below.
The term RCER for the Toroidal Cavity with built-in Elliptical Ring looks incorrect. May be it will be better to use the term TCER?
The main statement: “Since the wave length of the incident light is less than the thickness of the designed structure, the two-dimensional model can be used to approximate the three-dimensional model for calculation” looks ungrounded and must be proved by more arguments.
The most important situation do not belong to subject of the discussion.
For example, “The propagation properties of the designed structure are analyzed adopting finite element method (FEM) [24] and coupling mode theory (CMT)[25]. The transverse magnetic mode formula of the MIM waveguide is given below [27].” Namely, [24] never discuses the (FEM) and used Finite Difference Time Domain (FDTD) method and Circuit model based on transmission-line theory. [25] also uses FDTD but not the CMT. The formula of the MIM waveguide is never given in [27].
It must be proved the statement: “rectangular baffle…. guarantee that only transverse magnetic fields are induced in the integral structure”.
The method used for the simulation is almost uncertain for the reader and must be described in more details.
Discussion of the temperature sensor is omitted the data for FOM, which is very poor that making this sensor hardly has any practical use.
It must be given some arguments why does it need to implement the toroidal cavity with built-in elliptical ring. The comparison with the single ring structure is also needed.
In conclusion, this paper needs the major revision or the second resubmitting.
Round 2
Reviewer 2 Report
The paper has got some improvements and now may be suitable for the publication. Although I never believe in this structure as a temperature sensor due to a small FOM value but this design may be suitable for the other application and thus has to be published.